# Genome-wide CRISPR screen for HSV-1 host factors reveals PAPSS1 contributes to heparan sulfate synthesis

Takeshi Suzuki [1], Yoshitaka Sato [1,2 ✉], Yusuke Okuno[3], Fumi Goshima[1], Tadahisa Mikami[4], Miki Umeda[1], Takayuki Murata[1,5], Takahiro Watanabe[1], Koichi Watashi[6,7,8,9], Takaji Wakita[6], Hiroshi Kitagawa [4] & Hiroshi Kimura [1 ✉]

Herpes simplex virus type 1 (HSV-1) is a ubiquitous pathogen that causes various diseases in humans, ranging from common mucocutaneous lesions to severe life-threatening encephalitis. However, our understanding of the interaction between HSV-1 and human host factors remains incomplete. Here, to identify the host factors for HSV-1 infection, we performed a human genome-wide CRISPR screen using near-haploid HAP1 cells, in which gene knockout (KO) could be efficiently achieved. Along with several already known host factors, we identified 3'-phosphoadenosine 5'-phosphosulfate synthase 1 (*PAPSS1*) as a host factor for HSV-1 infection. The KO of *PAPSS1* in HAP1 cells reduced heparan sulfate (HepS) expression, consequently diminishing the binding of HSV-1 and several other HepS-dependent viruses (such as HSV-2, hepatitis B virus, and a human seasonal coronavirus). Hence, our findings provide further insights into the host factor requirements for HSV-1 infection and HepS biosynthesis.

[1] Department of Virology, Nagoya University Graduate School of Medicine, Nagoya 466-8550, Japan. [2] PRESTO, Japan Science and Technology Agency (JST), Kawaguchi 332-0012, Japan. [3] Department of Virology, Nagoya City University Graduate School of Medical Sciences, Nagoya 467-8601, Japan. [4] Laboratory of Biochemistry, Kobe Pharmaceutical University, Kobe 658-8558, Japan. [5] Department of Virology and Parasitology, Fujita Health University School of Medicine, Toyoake 470-1192, Japan. [6] Department of Virology II, National Institute of Infectious Diseases, Tokyo 162-8640, Japan. [7] Research Center for Drug and Vaccine Development, National Institute of Infectious Diseases, Tokyo 162-8640, Japan. [8] Department of Applied Biological Sciences, Tokyo University of Science, Noda 278-8510, Japan. [9] Institute for Frontier Life and Medical Sciences, Kyoto University, Kyoto 606-8507, Japan. ✉email: yssato@med.nagoya-u.ac.jp; hkimura@med.nagoya-u.ac.jp

Herpes simplex virus type 1 (HSV-1), a ubiquitous pathogen, causes various diseases in humans, ranging from the common mucocutaneous lesions to the severe life-threatening encephalitis[1]. Once infected, HSV-1 establishes latent infections in peripheral neurons and occasionally reactivates to cause recurrent lesions[2]. HSV-1 affects a broad range of hosts[3] and exclusively induces cell death via its cytopathic effect in vitro. It employs two modes to transmit from virus-producing cells to uninfected cells. Cell-to-cell infection occurs in the reactivation process. In this mode, viral particles transmit through a direct interaction between infected and neighboring uninfected cells[4]. In cell-free transmission, HSV-1 spreads to new hosts and often between cells within the same host, after which extracellular viral particles released from virus-producing cells enter the target cells. The attachment of the virus particle to the cell surface is the initial step in the cell-free infection of HSV-1. Evidence has demonstrated that HSV-1 glycoproteins (gC and/or gB) on the viral envelope interact with heparan sulfate (HepS) on the cell surface for attachment[5]. Subsequently, attachment to HepS enhances the interaction between gD and one of its receptors (i.e., nectin-1, HVEM, or 3-O-sulfated HepS)[6–8], inducing a conformational change in gD, and recruiting gB, gH and gL[9,10]. This tetrameric complex of glycoproteins mediates viral entry into the cells via receptor-mediated fusion between the viral envelope and the host cell[11,12]. Specifically, four viral glycoproteins (gB, gD, gH, and gL) are required and considered sufficient for viral entry into host cells[13]. However, the host factors involved in this process have not been fully understood.

HepS is a linear polysaccharide ubiquitously expressed on the cell surface and in the extracellular matrix. HepS chains covalently bound to proteins are known as HepS proteoglycans (HSPGs)[14]. These chains are heavily modified via sulfation at various positions on their sugar residues, imparting an overall high negative charge and creating binding sites for different molecules. Physiologically, HepS plays a vital role in regulating various cellular functions, such as cell growth, adhesion, angiogenesis, and blood coagulation, through interactions between HSPGs and their partner molecules[15]. Furthermore, HepS is utilized by various viruses for efficient infection, owing to its molecular diversity. In addition to HSV-1, studies have identified HepS as a factor that facilitates the binding and cell entry of numerous viruses, including HSV-2[16], human coronavirus (HCoV) OC43[17], severe acute respiratory syndrome coronavirus 2 (SARS-CoV-2)[17,18], Ebola virus[19], human immunodeficiency virus (HIV)[20], and hepatitis B virus (HBV)[21]. However, the factors involved in regulating HepS biosynthesis are currently not completely understood.

Therefore, in this study, we performed a genome-wide CRISPR screen for HSV-1 host factors and highlighted the importance of HepS in HSV-1 infection. As a result, we identified 3′-phosphoadenosine 5′-phosphosulfate synthase 1 (PAPSS1) as a critical factor for HSV-1 infection. The genetic ablation of PAPSS1 was sufficient to abolish HepS expression in HAP1 cells and consequently reduce the binding of various pathogenic viruses. Moreover, while the single KO of PAPSS1 slightly affected HepS biosynthesis in some PAPSS2-expressing cell lines, the double KO of PAPSS1 and PAPSS2 resulted in reduced HepS expression and higher resistance against HSV-1 infection in human retinal pigment epithelial-1 (RPE-1) cells, indicating a redundant role of PAPSS1 and PAPSS2 in HepS biosynthesis.

## Results

### A genome-wide CRISPR screen identifies host factors for HSV-1 infection.
To identify the genes required for HSV-1 infection, we performed a genome-wide CRISPR screen using a human near-haploid cell line (HAP1) (Fig. 1a). We used haploid cells because of their higher efficiency in generating loss-of-functional mutations relative to diploid cells[22]. First, we established HAP1 cells that stably expressed Cas9 (HAP1/Cas9). Then, we transduced these cells using a pooled lentivirus, encoding single-guide RNAs (sgRNAs) and targeting 19,052 genes (6 sgRNA/gene), 1864 microRNA (miRNA) (4 sgRNA/miRNA), and 1000 non-targeting control sgRNAs at a multiplicity of infection (MOI) of 0.3. The transduction step ensured that each cell generally expressed only one sgRNA. After the mutagenesis by lentiviral transduction and antibiotic selection for a week, we infected 20 million cells with HSV-1 at an MOI of 0.1 and selected the surviving cells. Subsequently, the survivor cells were subjected to a second challenge with HSV-1 at two weeks after the first HSV-1 challenge, to enable the enrichment of cells resistant to HSV-1 infection. The cells were expanded further for 2 weeks, and the genomic DNA of the survivor cells was extracted and subjected to sequencing. As shown in Fig. 1b, sgRNAs in the virus-infected group were selected from <3% of the total sgRNAs in the duplicate screening step, and then each gene was ranked as either the most enriched sgRNA or the second-most enriched sgRNA[23] (Fig. 1c). The known HSV-1 entry receptor NECTIN-1, which interacts with HSV-1 glycoprotein D[6], was ranked at the top in our enrichment analysis, demonstrating the technical quality of the screening process. Additionally, seven genes in the HepS biosynthesis pathway (XYLT2, B4GALT7, B3GAT3, EXTL3, EXT1, EXT2, and SLC35B2) were markedly enriched (Fig. 1c). While these six former genes encode enzymes that catalyze the HepS backbone formation[14] (Fig. 1d), SLC35B2 serves as the transporter of adenosine 3′-phospho 5′-phosphosulfate (PAPS), the universal sulfuryl donor for sulfation. PAPS was previously reported to be involved in HepS biosynthesis[24,25]. We focused on the genes supported by ≥2 sgRNAs with >150 RPM for further validations since a sgRNA targeting EXT2 showed the lowest abundance (150.5 RPM) among these HepS-related genes (Fig. 1c; Supplementary Table 1). In addition to NECTIN1 and HepS-related genes, our screen identified three candidate genes (i.e., IRF2BPL, PAPSS1, and VANGL2) and a candidate miRNA (MIR4647) (Fig. 1c).

We performed validation experiments for NECTIN1, several HepS-related genes (including XYLT2 and EXT2), and three additional candidate genes. Among the HepS-related genes, we selected XYLT2 and EXT2 for validation since XYLT2 initiates the biosynthesis of glycosaminoglycan chains[26], and EXT2 is involved in the later step of chain formation[27] (Fig. 1d). Then, we generated one or two HAP1 knockout (KO) clones for each gene via the CRISPR-Cas9 approach using several sgRNAs (Supplementary Fig. 1a). Later, these clones were infected with HSV-1, and their viabilities were measured at 48 h post-infection (hpi). Results showed that the cytopathic effects of HSV-1 infection were significantly suppressed in the KO clones of NECTIN1, XYLT2, EXT2, and PAPSS1 compared with the wild type (WT) and nontargeting control cells (Fig. 1e). We also confirmed this validation through HSV-1 infection at a different MOI (Supplementary Fig. 1b). We observed that the perturbation of MIR4647 by a miRNA inhibitor did not suppress the cytopathic effects of HSV-1 infection (Supplementary Fig. 2). Of note, the genomic locus of MIR4647 is overlapped in the 3′-UTR of SLC35B2[28], suggesting the possibility that sgRNAs targeting MIR4647 reduce the expression of SLC35B2.

Therefore, our screening process collectively identified NECTIN1, PAPSS1, and a series of genes in the HepS biosynthesis pathway as important for HSV-1 infection.

### XYLT2 and EXT2 enhance the binding of HSV-1 to the cell surface through HepS expression.
Next, to investigate the role of genes that catalyze the sugar chain formation of HepS in HSV-1

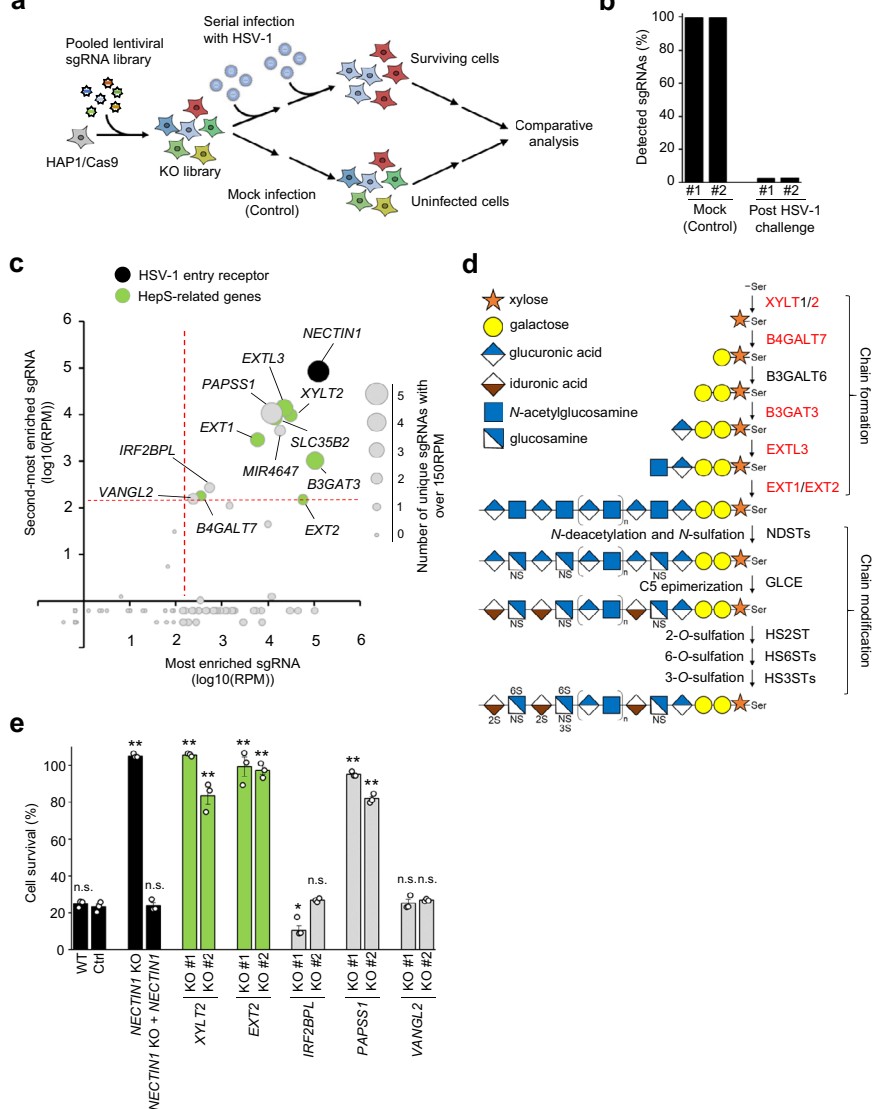

**Fig. 1 Genome-wide CRISPR screen to identify genes involved in HSV-1 infection. a** Schematic representation of the CRISPR-based KO screening process. HAP1/Cas9 cells were mutagenized with a pooled lentiviral single-guide RNAs (sgRNA) library containing 121,449 unique sgRNAs targeting 19,052 genes and 1864 miRNAs. Mutant cells were infected with HSV-1, following which the viral infection-resistant cells were harvested. The control groups were mock-infected with PBS and cultured for the same period. The abundance of each sgRNA in the control and selected groups was determined through next-generation sequencing. This screening was performed in duplicate at an estimated coverage of 150 cells/sgRNA. **b** Abundance of sgRNAs in the mock-infected control and HSV-1-infected groups. **c** Scatter plot showing the most enriched sgRNA (log-transformed RPM: X-axis) versus the second-most enriched sgRNA (log-transformed RPM: Y-axis) for each gene following selection by HSV-1 infection. The size of the plots indicates the number of unique sgRNA detected with >150 RPM. Genes supported by ≥2sgRNAs with >150 RPM (red lines) were selected for further validation. **d** Overview of HepS biosynthesis. Genes supported by two or more sgRNA with over 150 RPM on our screen are highlighted in red font. **e** Validation of candidate genes. One or two KO HAP1 clones of the indicated genes were obtained by the CRISPR-Cas9 system. The wildtype (WT), nontargeting control (Ctrl), and KO clones were infected with HSV-1at a MOI of 3, and the cell viabilities were measured at 48 hpi via the MTS assay. The results are presented as means ± SEM of three independent experiments. Asterisks, $p < 0.05$; Double asterisks, $p < 0.01$; n.s. not significant.

infection, two KO clones of *XYLT2* and *EXT2* each and their rescued clones were obtained using HAP1 cells (Fig. 2a). Fluorescence-activated cell sorting (FACS) analysis with an anti-HepS antibody revealed that HepS expression was downregulated to background levels in both the *XYLT2*- and *EXT2*-KO clones and the same extent in the heparinase-treated cells (Figs. 2b, c). Furthermore, treatment with an antagonist to HepS, Surfen, inhibited HSV-1-infection-mediated cell death (Fig. 2d). Although the *XYLT2*- and *EXT2*-KO clones demonstrated higher survivability against HSV-1 infection than the control clone at 48 hpi (Fig. 2e), the complementation of the lost gene canceled the

resistance (Fig. 2e). Consistent with these findings, progeny virus production was also reduced in the KO clones by 1–2 log fold compared with the control clone (Fig. 2f). Subsequently, since HepS involves the binding of HSV-1 on the cell surface[12], we performed a binding assay using *XYLT2*- and *EXT2*-KO HAP1 cells. The levels of viral DNA on the cell surface were dramatically decreased by 30–200 fold in both *XYLT2*- and *EXT2*-KO clones (Fig. 2g). To further investigate the roles of these genes in HSV-1 infection using epithelial cells, which are HSV-1 natural targets, we obtained *XYLT2*- and *EXT2*-KO clones using RPE-1 cells (Supplementary Fig. 3a). Similar to HAP1 cells,

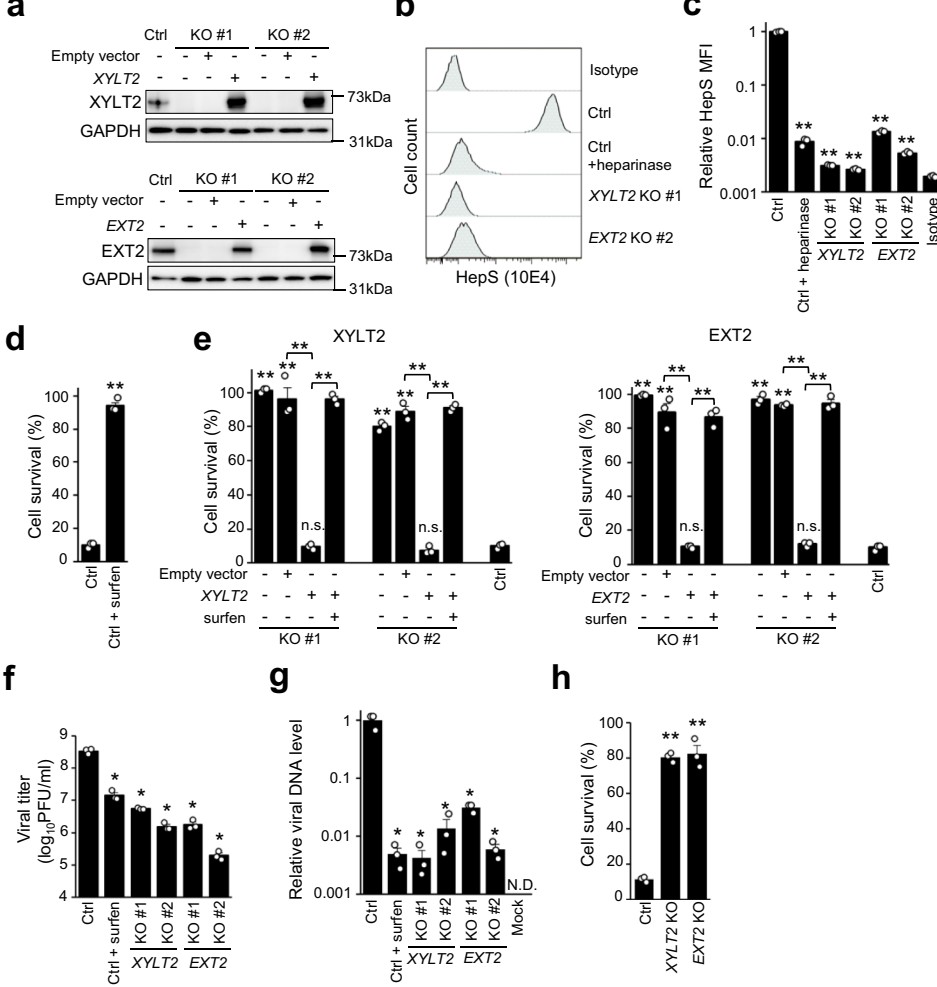

**Fig. 2 XYLT2 and EXT2 as essential genes for HSV-1 binding and infection. a** Western blotting analysis confirming the absence or complementation of *XYLT2*, the *EXT2* expression in KO, and rescued clones of HAP1 cells. **b**, **c** HepS expression analysis by FACS. After treatment with or without heparinase for 1 h at 37 °C, the cells were stained using an anti-HepS antibody. The graph summarized the mean fluorescent intensity (MFI) of the HepS expression analysis of three independent experiments (**c**). Results are presented as means ± SEM. Viability of the nontargeting control clone (**d**) and the *XYLT2-* and *EXT2*-KO clones (**e**) of HAP1 cells after HSV-1 infection. The cells were either pretreated with 5 µM surfen hydrate or DMSO and infected with HSV-1 at a MOI of 9. The cell viabilities were measured via the MTS assay at 48 hpi. The results are presented as means ± SEM of three independent experiments. **f** Progeny virus production of HSV-1 in the control, *XYLT2*-KO, and *EXT2*-KO clones of HAP1 cells. Following surfen hydrate or DMSO treatment, the cells were infected with HSV-1 at a MOI of 1. The progeny viruses were collected at 48 hpi and titrated via the plaque assay. The results are presented as means ± SEM of three independent experiments. **g** Binding of HSV-1 in the control, *XYLT2*-KO, and *EXT2*-KO clones of HAP1 cells. After surfen hydrate or DMSO treatment, the cells were adsorbed with HSV-1 at an MOI of 50 for 1 h and 4 °C. The nontargeting control cells that were mock-infected with PBS were used as a negative control. After removing the residual viruses, the viral DNA was extracted and quantified using real-time PCR. The results are presented as means ± SEM of three independent experiments and shown as the relative DNA level of the control cells. **h** Viability of the nontargeting control clone, *XYLT2*-KO, and *EXT2*-KO human retinal pigment epithelial-1 cell clones after HSV-1 infection. The cells were infected with HSV-1 at an MOI of 9. Then, the viabilities were measured via the MTS assay at 48 hpi. The results are presented as means ± SEM of three independent experiments. Asterisks, *p* < 0.05; Double asterisks, *p* < 0.01; n.s. not significant; N.D. not detected.

*XYLT2-* and *EXT2*-KO RPE-1 cells showed reduced HepS expression and higher resistance against HSV-1 infections than the control clone (Fig. 2h and Supplementary Fig. 3b, c). These findings confirmed that HepS enhances the attachment of HSV-1 particles on the cell surface, assisting the cell-free infection of HSV-1.

**HepS is important for HSV-1 cell-to-cell infection.** In addition to cell-free infections, we investigated whether HepS contributed to the other modes of HSV-1 transmission, i.e., cell-to-cell infection. *XYLT2* initiates sugar chain formation by transferring one xylose to the serine residues of core proteins, followed by the

conjugation of two galactose molecules and one glucuronic acid (Fig. 1d). These first four sugars are shared by other glycosaminoglycans: chondroitin sulfate and dermatan sulfate[26]. Therefore, to ensure that only HepS expression was affected, an *EXT2*-KO clone and its rescue clone in RPE-1 cells were generated using the CRISPR-Cas9 system (Fig. 3a). Here we employed RPE-1 cells to accurately calculate the plaque areas because the plaque formation in HAP1 cells by HSV-1 infections showed an obscure margin. The reduced HepS expression of *EXT2*-KO RPE-1 cells was recovered by complementing with *EXT2* (Figs. 3b, c). Subsequently, RPE-1 WT cells infected with HSV-1 were used as an inoculate, after which these inoculates (30 cells/well) were cocultured with a pre-seeded *EXT2*-KO clone or nontargeting

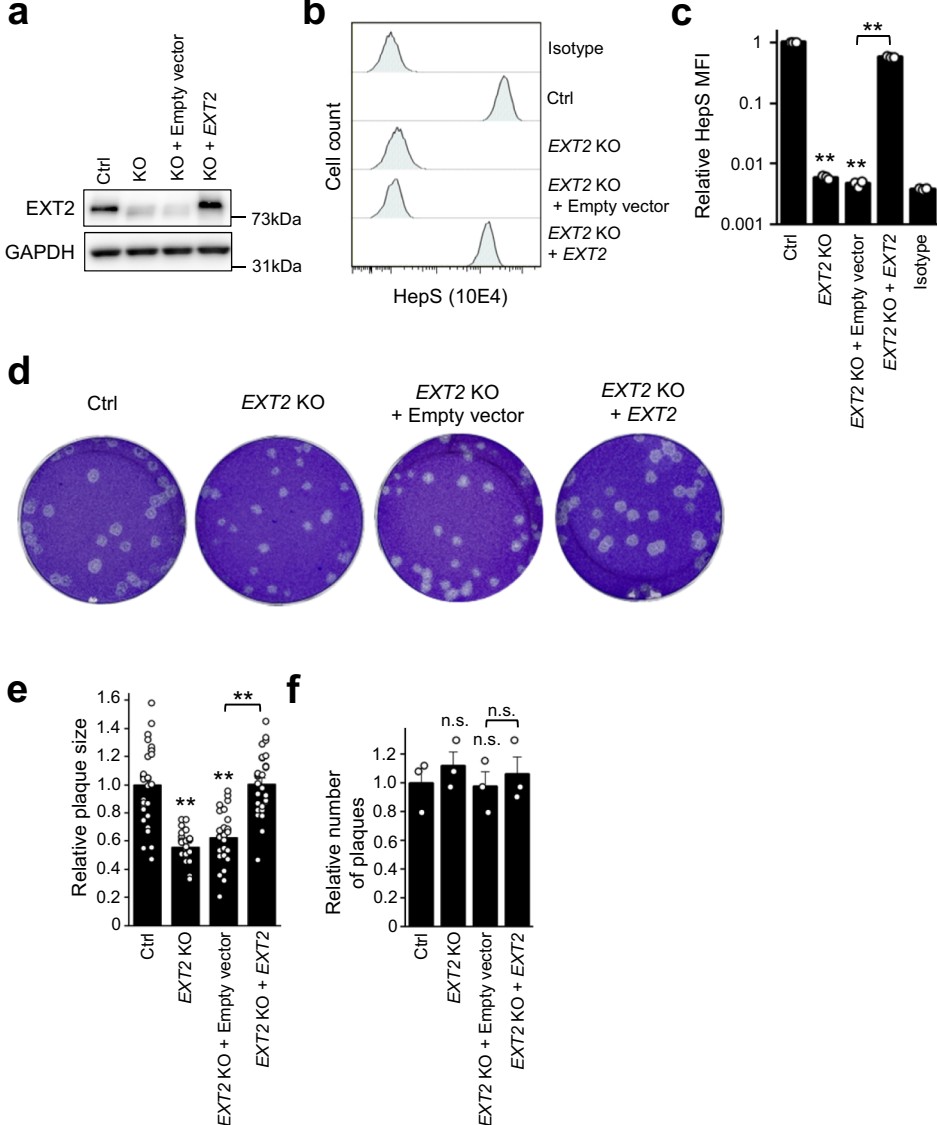

**Fig. 3 HepS is important for HSV-1 cell-to-cell infection. a** Western blotting confirming the absence or complementation of *EXT2* expression in the KO and complemented clones of RPE-1 cells. **b**, **c** HepS expression analysis by FACS. The graph summarizes the MFI of the HepS expression analysis of three independent experiments (**c**). The results are presented as means ± SEM. **e–f** HSV-1 cell-to-cell spreading assay. Each clone was inoculated with 30 cells/ well of wildtype RPE-1, which pre-infected with HSV-1 and then incubated in a media containing 5 mg/mL pooled human IgG (**d**). The areas of 10 individual plaques were measured (**e**), and the number of plaques was counted (**f**). The results are presented as means ± SEM of three independent experiments and shown as the relative size (**e**) and the relative number of plaques (**f**). Double asterisks, $p < 0.01$; n.s. not significant.

control clone. Neutralizing antibodies against HSV-1 were added to the media to prevent cell-free spread. The plaque sizes for the *EXT2*-KO clone were decreased compared with that of the non-targeting control clone at 72 hpi (Figs. 3d, e), whereas the number of plaques was almost the same (Figs. 3d–f). Furthermore, the complementation of *EXT2* recovered the plaque size (Figs. 3d, e). These findings propose that HepS contributes to both the cell-free and cell-to-cell infection of HSV-1.

**Knockout of *PAPSS1* in HAP1 cells impairs the biosynthesis of HepS and reduces the attachment of HSV-1 on the cell surface.** Our screening identified *PAPSS1* as a host factor of HSV-1 infection (Fig. 1). *PAPSS1* encodes an enzyme that synthesizes PAPS. PAPS is synthesized by PAPSS1 in the cytosol or nucleus, then transported into the lumen of the Golgi apparatus, where sulfation of various substrates, including HepS, occurs[29,30]

(Fig. 4a). The transporter of PAPS is encoded by *SLC35B2*, which was also highly enriched in our screen (Fig. 1), and it was pre-viously considered essential in HepS biosynthesis in HAP1 cells and 293 T cells[24,25]. Thus, we hypothesized that *PAPSS1* is also required for HepS biosynthesis. To test this, we obtained two *PAPSS1*-KO clones of HAP1 cells and the rescued clones (Fig. 4b). We analyzed HepS expression by FACS analysis. As shown in Figs. 4c, d, FACS analyses revealed drastically dimin-ished background-level HepS expression in *PAPSS1*-KO clones, which was rescued by the complementation of *PAPSS1*. Fur-thermore, HepS chain analysis by high-performance liquid chromatography (HPLC) demonstrated that although the total amount of HepS disaccharides was reduced by approximately 30%, the sulfation degree was decreased approximately by 20-fold in *PAPSS1*-KO cells (Fig. 4e, Supplementary Fig. 4, and Supple-mentary Table 2). Consequently, *PAPSS1*-KO cells exhibited a robust resistance against HSV-1 infection, presenting a >90%

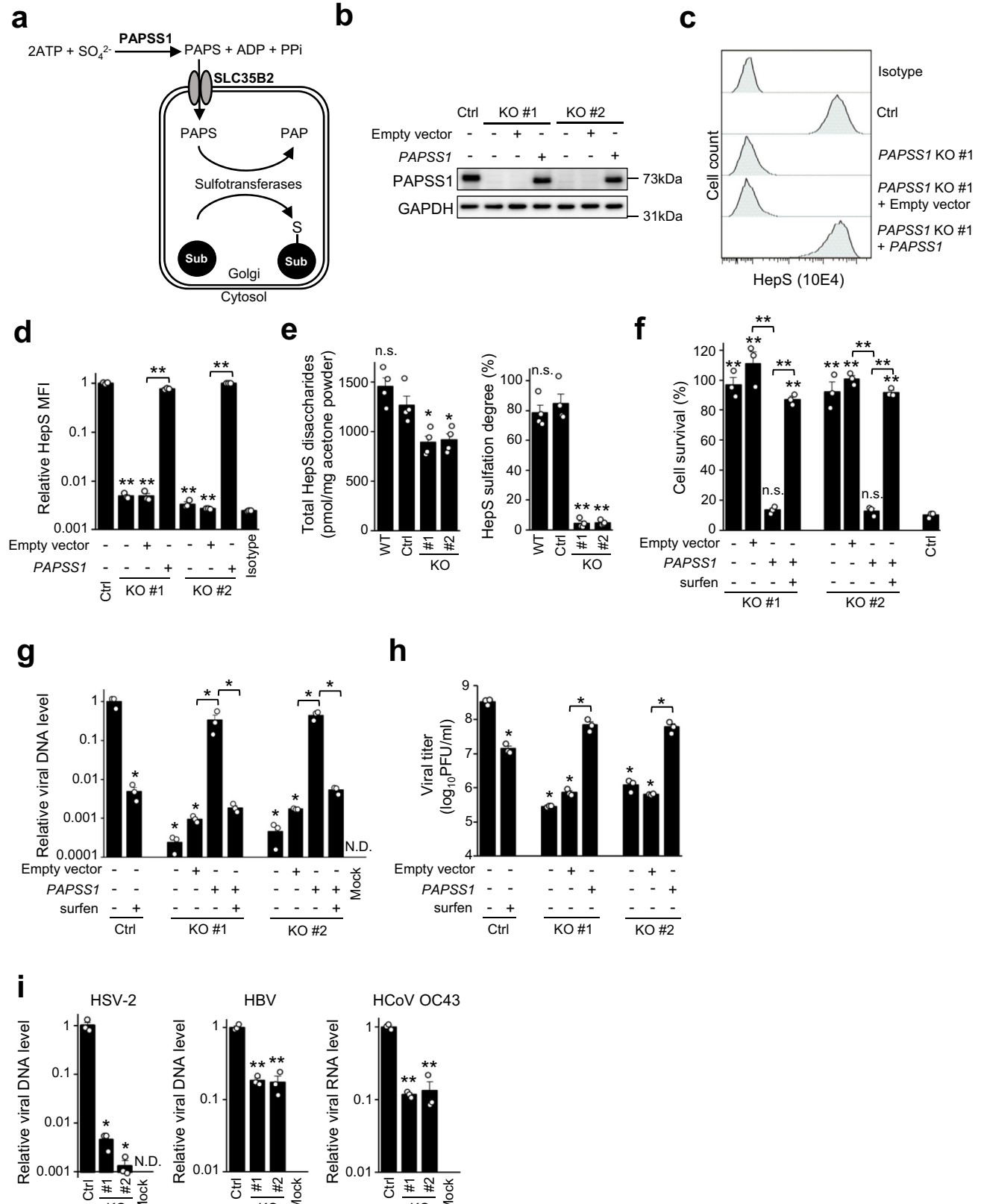

survival rate at 48 hpi (Fig. 4f). The knockout of *PAPSS1* decreased the binding of virus particles on the cell surface (Fig. 4g), causing a reduction in progeny virus production by 2–3 log fold (Fig. 4h).

We further examined whether *PAPSS1* contributes to the binding of other viruses that are already known to use HepS for binding[16,17,21]. Likewise, for HSV-1, the binding of HSV-2, HBV, and HCoV OC43 was significantly reduced in the *PAPSS1*-KO clones compared with the control cells (Fig. 4i). Hence, our cumulative findings suggested that *PAPSS1* influences HepS biosynthesis, contributing to viral infections by assisting the binding of virus particles.

**Fig. 4 *PAPSS1* is an important gene for HepS biosynthesis. a** Schematic representation showing the contribution of *PAPSS1* and *SLC35B2* to the sulfation of substrates within the Golgi apparatus. **b** Western blotting confirming the absence or complementation of *PAPSS1* expression in the KO and rescued clones of HAP1 cells. **c**, **d** HepS expression analysis by FACS. The graph summarizes the MFI of HepS expression analysis of three independent experiments (**d**). The results are presented as means ± SEM. **e** HepS disaccharide analysis by HPLC. The total amount and the sulfation degree of HepS disaccharides of the WT, nontargeting Ctrl, and *PAPSS1*-KO clones of HAP1 cells were measured. The results are presented as means ± SEM of four independent experiments. **f** Viability of *PAPSS1*-KO clones of HAP1 cells after HSV-1 infection. The cells were infected with HSV-1 at a MOI of 9. Their cell viabilities were measured using the MTS assay at 48 hpi. The results are presented as means ± SEM of three independent experiments. **g** Binding of HSV-1 in the *PAPSS1*-KO clones. The cells were adsorbed with HSV-1 at an MOI of 50 for 1 h at 4 °C. The nontargeting control cells that were mock-infected with PBS were used as a negative control. After removing the residual viruses, the viral DNA was extracted and quantified using qPCR. The results are presented as means ± SEM of three independent experiments and are shown as the relative DNA level compared with that of the control. **h** Progeny virus production of HSV-1 in the *PAPSS1*-KO clones. The cells were infected with HSV-1 at an MOI of 1. The progeny viruses were collected at 48 hpi and titrated using the plaque assay. The results are presented as means ± SEM of three independent experiments. **i** Binding of the indicated viruses in the control and the *PAPSS1*-KO clones. The cells were adsorbed with the virus for 1 h at 4 °C. The non-targeting control cells that were mock-infected with PBS were used as a negative control. After removing the residual viruses, the viral DNA (HSV-2 and hepatitis B virus) or RNA (human coronavirus OC43) was extracted. The viral genome was quantified using qPCR. The results are presented as means ± SEM of three independent experiments and shown as relative DNA levels compared with that of the control. Asterisks, $p < 0.05$; Double asterisks, $p < 0.01$; N.D. not detected; n.s. not significant.

**PAPSS1 influence on HepS biosynthesis depends on PAPSS2 expression**. We further investigated whether *PAPSS1* was necessary for HSV-1 infection and HepS biosynthesis in RPE-1 cells. As shown in Fig. 5a, *PAPSS 1*-KO RPE-1 cells showed significantly, but weaker resistance against HSV-1 infection than *XYLT2*- and *EXT2*-KO clones (Fig. 2h). Consistent with this, FACS analysis presented a slight reduction in the HepS expression of *PAPSS1*-KO RPE-1 cell clones (Figs. 5b, c). To evaluate the impact of *PAPSS1*-KO on HepS biosynthesis in other cell lines, we additionally obtained *PAPSS1*-KO clones using human gastric adenocarcinoma (AGS) cells and lung adenocarcinoma A549 cells (Supplementary Fig. 5). The *PAPSS1*-KO clones of AGS cells showed reduced HepS expression to the background level, similar to HAP1 cells. However, the *PAPSS1*-KO clones of A549 cells showed a limited reduction in HepS expression, similar to RPE-1 cells (Figs. 5b, c). This limited effect of *PAPSS1*-KO on HepS biosynthesis implied the expression of a paralog enzyme, *PAPSS2*. When we investigated the level of *PAPSS2* expression, it correlated with the limited reduction of HepS expression by *PAPSS1*-KO (Fig. 5d). To assess the functional redundancy between *PAPSS1* and *PAPSS2*, we generated *PAPSS2*-KO clones and double KO (DKO) clones of *PAPSS1* and *PAPSS2* using RPE-1 cells (Fig. 5e). Similar to single *PAPSS1*-KO clones, single *PAPSS2*-KO clones showed a limited reduction of HepS expression in RPE-1 cells and little or no gain of resistance to HSV-1 infection. However, DKO clones showed a strong decrease in HepS expression and robust resistance to HSV-1 infections (Figs. 5f–h). These findings propose the redundant role of *PAPSS1* and *PAPSS2* in the HepS biosynthesis of RPE-1 cells.

## Discussion

Loss-of-function genetic screening has identified various host factors required for viral infection[31,32]. This screening strategy involves the isolation of mutations that render host cells resistant to viral infection using a knockdown or knockout mutant library. Compared with previous gene knockdown approaches, such as RNAi, the complete ablation of gene expression using a haploid screen or CRISPR screen produces clearer phenotypes on virus infection, enabling the identification of critical host factors for viral infection. This approach has been applied previously to uncover host factors involved in viral entry. Haploid screens via insertional mutagenesis discovered essential receptors for the Ebola virus[33] and Lassa virus[34,35]. Recently, CRISPR screening identified host factors that regulate SARS-CoV-2[17,36,37], common cold coronaviruses[17,37], HIV[38], adeno-associated virus[39], flavivirus[40], and Zika virus[41]. In addition to identifying the entry receptor of the Lassa virus, Jae et al. revealed the gene networks encoding for enzymes and accessory factors involved in the O-linked glycosylation of α-dystroglycan[35]. Likewise, we also performed a genome-wide CRISPR screen for HSV-1 host factors and revealed a set of genes involved in HepS biosynthesis, including a viral entry receptor, *NECTIN1*. The biosynthesis of HepS has also been reported as a sequential multistep process that occurs in the Golgi apparatus (Fig. 1d)[42]. Our results showed that knocking out *XYLT2* and *EXT2* caused a severe defect in HepS expression on the cell surface (Fig. 2). *XYLT2* is a glycosyltransferase that initiates the biosynthesis of glycosaminoglycan chains (both HepS and chondroitin sulfate/dermatan sulfate) in proteoglycans[26], and *EXT2* is involved in heparan sulfate chain elongation[27]. In contrast to the genes related to the backbone formation of HepS, no other sulfotransferases were enriched in our screening process, possibly because of their functional redundancy.

Additionally, all the host factors identified in our screening were involved in viral attachment and entry, the earliest stage of viral infection. This bias might be caused by the long selection period of our screening. The cells lacking host factors engaged in the later stages of infections would not allow viruses to replicate or mature, but allow to enter the cell. The cytopathic and cytocidal effects are caused even by the entry of UV-inactivated HSV-1[43,44] and replication-defective mutants[45]. Alterenatively, the cells lacking host factors engaged in the later stages of infections could not propagate sufficiently or underwent apoptosis for harboring viruses[46,47]. Therefore, the cells allowed viruses to enter the cell might be eliminated during the selection.

Although we demonstrated that HepS contributed to both the cell-free and cell-to-cell infection of HSV-1 (Figs. 2 and 3), how HepS is involved in the cell-to-cell infection of HSV-1 remains unclear. During vaccinia virus infection, the viral factor, F11L, and a vaccinia growth factor contribute to the cell-to-cell spread of the vaccinia virus via the activation of RhoA- and EGFR-mediated cell motility, respectively[48–50]. Cell motility is essential for the dissemination of various viruses[48,51–53]. Recently, it was reported that the interaction between HSV-1 glycoprotein E and prohibin-1 activates the MAPK/ERK pathway, promoting cell-to-cell spread[54]. This signaling pathway also promotes cell motility[55,56]. Similarly, HepS on the cell surface can serve as an anchor to facilitate the endocytosis of growth factors, subsequently inducing rapid and efficient motility[57–59]. These findings invoke a model in which HSV-1 uses HepS to promote cell motility, increasing cell-to-cell contact with the neighboring uninfected cells. However, further studies are warranted to determine how HepS contributes to HSV-1 cell-to-cell transmission, for instance, whether the motility of the infected cells is

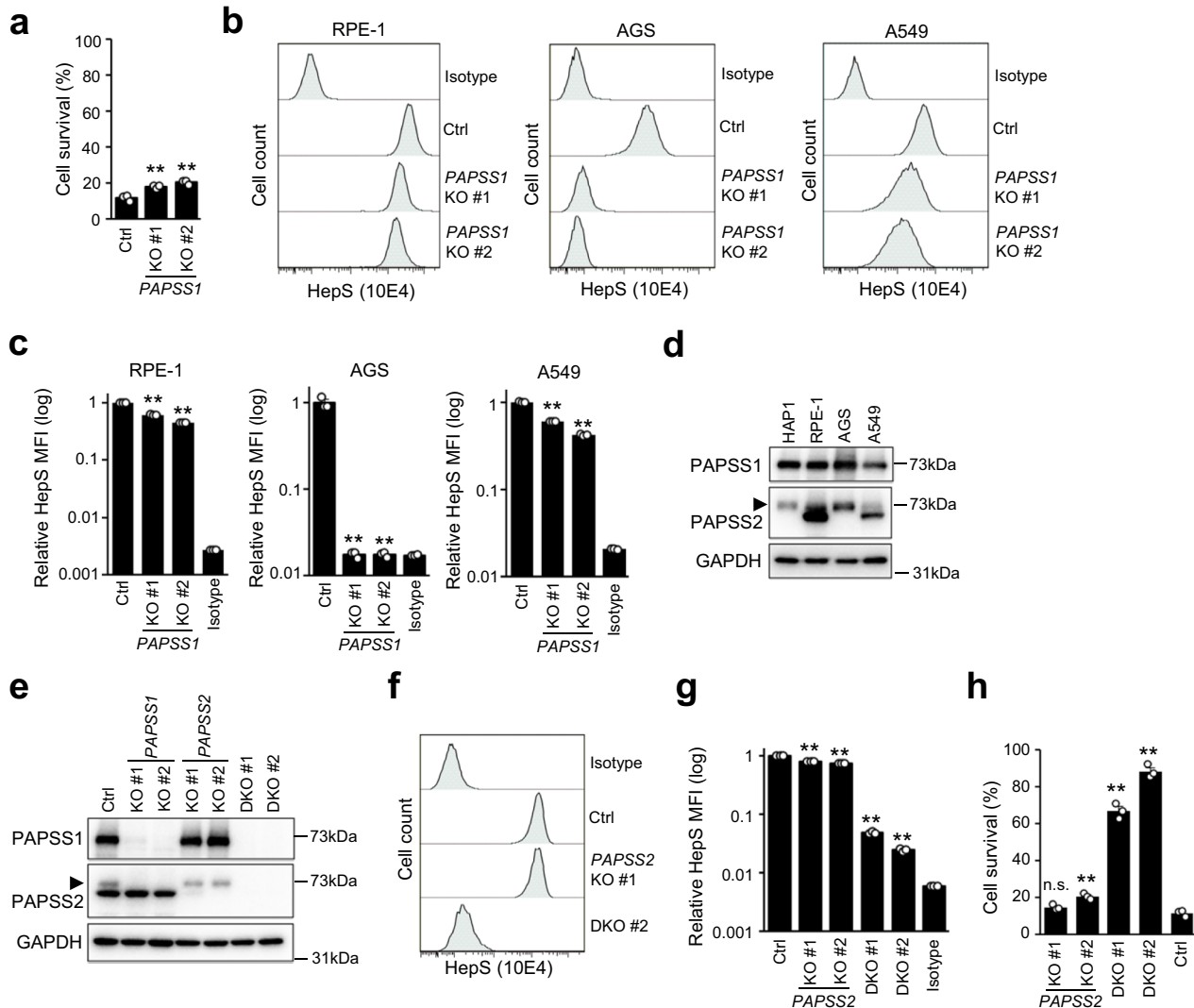

**Fig. 5 *PAPSS1*-dependency of HepS biosynthesis differs between cell lines due to the *PAPSS2* expression. a** Cell viability of *PAPSS1*- KO of RPE-1 cell clones after HSV-1 infection. The cells were infected with HSV-1 at a MOI of 9. Their cell viabilities were measured using the MTS assay at 48 hpi. The results are presented as means ± SEM of three independent experiments. **b**, **c** HepS expression analysis of *PAPSS1*-KO clones of RPE-1, AGS, and A549 cells by FACS (**b**). The graph summarizes the MFI of the HepS expression analysis of three independent experiments (**c**). The results are presented as means ± SEM. **d** Western blotting showing the expression of *PAPSS1* and *PAPSS2* in HAP1, RPE-1, AGS, and A549 cells. The arrowhead indicates the cross-reactive band of PAPSS1. **e** Western blotting confirming the absence of *PAPSS1* and/or *PAPSS2* expression in the KO clones of RPE-1 cells. The arrowhead indicates the cross-reactive band of PAPSS1. **f**, **g** HepS expression analysis of *PAPSS2*-KO and *PAPSS1*- and *PAPSS2*-double KO (DKO) clones of RPE-1 cells by FACS (**f**). The graph summarized the MFI of HepS expression analysis of three independent experiments (**g**). The results are presented as means ± SEM. **h** Viability of *PAPSS2*-KO and DKO clones of RPE-1 cells after HSV-1 infection. The cells were infected with HSV-1 at an MOI of 9. The cell viabilities were measured by the MTS assay at 48 hpi. The results are presented as means ± SEM of three independent experiments. Double asterisks, *p* < 0.01; n.s. not significant.

affected by HepS expression on their cell surface and/or by the MAPK/ERK pathway via HepS-dependent endocytosis.

The HepS chain's modification is complex, occurring via sulfation. As a result, its molecular diversity generates binding sites for multiple protein ligands[60,61]. This study identified *PAPSS1* as a critical factor of HepS biosynthesis in HAP1 cells (Fig. 4). *PAPSS1* produces the universal sulfate donor PAPS required for the sulfation of polysaccharides. We demonstrated a diminished HepS expression of *PAPSS1*-KO clones through FACS analysis. Consistent with this, HepS disaccharide analysis also revealed a decrease in the level of HepS disaccharides and entire sulfation, resulting in reduced viral attachment on the cell surface in HAP1 cells (Fig. 4). In humans, PAPS, the sole source of sulfate, is synthesized from adenosine 5-prime triphosphate (ATP) and

inorganic sulfate by two enzymes; *PAPSS1* and *PAPSS2*. While *PAPSS1* is ubiquitously expressed in human adult tissues, *PAPSS2* shows a more restricted expression pattern[62,63]. In this study, KO analysis of these enzymes showed *PAPSS1* and *PAPSS2* functions to be redundant during HepS biosynthesis in RPE-1 cells (Fig. 5). In a previous study, defects in *PAPSS2* caused the Pakistani type of spondyloepimetaphyseal dysplasia[62,64]. However, no information is available on why the ubiquitously expressed *PAPSS1* cannot compensate for the loss of *PAPSS2* activity. Differences in subcellular compartmentalization[30], catalytic efficiency, and expression patterns during cartilage and bone development[62,63] might explain the differential effects of *PAPSS1* and *PAPSS2*.

In summary, our study findings confirmed the significance of HepS in HSV-1 infection and identified *PAPSS1* as the key factor

for HepS biosynthesis. The CRISPR-mediated inactivation of *PAPSS1* abolished the sulfated HepS expression on the cell surface, which resulted in the reduced binding of various pathogenic viruses in HAP1 cells. Furthermore, in some other cells that express *PAPSS2*, such as RPE-1 and A549 cells, *PAPSS2* compensated for *PAPSS1* in HepS biosynthesis. Therefore, our study provides further insights into the host factors required for HSV-1 infection and HepS biosynthesis regulation.

## Methods

**Cells and virus stocks**. HAP1 cells were purchased from Horizon (Cambridge, UK) and cultured in Iscove's Modified Dulbecco's Medium (IMDM) (Nakalai Tesque, Kyoto, Japan) supplemented with 10% fetal bovine serum (FBS), 1% GlutaMAX (Thermo, Waltham, MA, USA), and 1% penicillin-streptomycin (Pen/St) (Thermo). HEK293T (CRL-3216; ATCC), A549 (CCL-185; ATCC), and Huh7 (RCB 1942; RIKEN BRC Cell Bank) cells were cultured in Dulbecco's modified Eagle's medium (DMEM) (Sigma-Aldrich, St. Louis, MO, USA) supplemented with 10% FBS, 1% GlutaMAX, and 1% Pen/St. RPE-1 cells (CRL-4000; ATCC) were cultured in DMEM/F-12 medium (Nakarai Tesque) supplemented with 10% FBS, 1% GlutaMAX, and 1% Pen/St. Vero cells (CCL-81; ATCC) were cultured in Eagle's minimal essential medium supplemented with 10% calf serum (CS), 1% L-glutamine (Thermo), and 1% Pen/St. AGS cells (CRL-1739; ATCC) were cultured in RPMI-1640 medium (Nakalai Tesque) supplemented with 10% FBS, 1% GlutaMAX, and 1% Pen/St. HepAD38 cells (a gift from Christoph Seeger) were cultured as described previously[65]. All cells were maintained at 37 °C and 5% $CO_2$ unless otherwise indicated.

HSV-1 strain F and HSV-2 strain 186 were propagated in Vero cells. HCoV strain, OC43, was obtained from ATCC (VR-1558) and propagated in Huh7 cells at 35 °C. HBV genotype D (subtype ayw) was prepared from HepAD38 cells, as described previously[65].

**Antibodies, plasmids, and reagents**. Anti-HepS (10E4 epitope; H1890, 1:50) mouse monoclonal antibody was purchased from US Biological Life Sciences (Salem, MA, USA). Anti-XYLT2 (G-1; sc-374134, 1:500), anti-EXT2 (A-2; sc-514092, 1:500), anti-PAPSS1 (A-2; sc-376244, 1:500), anti-PAPSS2 (SQ-19; sc-100801, 1:500) and normal mouse IgM (sc-3881, 1:50) antibodies were obtained from Santa Cruz Biotechnology (Dallas, TX, USA). Anti-IRF2BPL (NBP2–56241, 1:500) antibody was purchased from Novus Biologicals (Littleton, CO, USA). Anti-VANGL2 (clone 2G4; MABN750, 1:500) antibody was obtained from Merck (Darmstadt, Germany). Anti-GAPDH (D16H11; #5174, 1:2000), horseradish peroxidase-conjugated anti-mouse or anti-rabbit secondary antibodies (#7074 and #7076, 1:1000 and 1:1000) were purchased from Cell Signaling Technology (Danvers, MA, USA). A horseradish peroxidase-conjugated anti-rat (SA00001-15, 1:1000) secondary antibody was obtained from Proteintech Group (Rosemont, IL, USA). Alexa Fluor 488-conjugated anti-mouse IgM secondary antibody (A-10680, 1:50) was obtained from Thermo.

The full-length cDNAs of *EXT2*, *NECTIN1*, *PAPSS1*, and *XYLT2* were obtained by reverse transcriptase-PCR (RT-PCR) and cloned into the CSII-CMV-MCS-IRES2-Bsd vector (RIKEN BioResource Center, Wako, Japan) using the In-Fusion cloning system (Takara Bio, Kusatsu, Japan). The primers used for RT-PCR are listed in Supplementary Table 3. To KO the gene of interest, sgRNA was designed using CHOPCHOP (https://chopchop.cbu.uib.no)[66] and cloned into px459 (a gift from Feng Zhang; #48139, Addgene, Watertown, MA, USA) or lentiviral expression constructs containing Cas9 and blasticidin- or neomycin-resistant genes, generated by VectorBuilder® (Shenandoah, TX, USA). The oligos of the used sgRNAs are shown in Supplementary Table 4. Inserted DNA sequences of each vector were confirmed through direct DNA sequencing.

Surfen hydrate (S6951) and heparinase II (#P0736) were purchased from Sigma-Aldrich and New England Biolabs (Ipswich, MA, USA), respectively. miRNA inhibitors (SMI-001-MIMAT0019709 and SMC-2101) were purchased from Bioneer Corporation (Daejeon, Republic of Korea). Lipofectamine 2000 was obtained from Thermo.

**Genome-wide CRISPR screen**. HAP1/Cas9 cells were generated through lentiviral transduction with lentiCas9-blast (a gift from Feng Zhang; #52962, Addgene), followed by antibiotic selection with blasticidin for seven days. To establish CRISPR KO libraries, 120 million HAP1/Cas9 cells were transduced with each lentiviral human GeCKO v2 library A and B (SureGuide GeCKO v2.0 Human Exome CRISPR Library; Agilent Technologies, West Cedar Creek, TX, USA) at an MOI of 0.3. Cells were subsequently selected using puromycin and cultured for 7 days. This culturing equates to >500-fold coverage of the library after selection. Twenty million selected cells (estimated coverage: 150 cells/sgRNA) were subjected to the first HSV challenge at an MOI of 0.1 with 5 mg/mL of pooled human IgG (Equitech-Bio, Kerrville, TX, USA). Survivor cells were cultured for 2 weeks, followed by a second HSV challenge at an MOI of 0.1 without IgG. After an additional 2 weeks, 20 million viable cells were harvested, followed by gDNA extraction with the QIAmp DNA Mini Kit (QIAGEN, Hilden, Germany). To ensure that HSV-1 attacked all the cells during the selection period, the same number of HAP1/Cas9 cells were infected with HSV-1 and cultured

likewise, and the survival of none of the cells was confirmed. The control group was mock-infected with PBS and cultured for the same period as the HSV-challenged group. To evaluate sgRNA enrichment, regions containing sgRNA cassette were amplified using the PrimeSTAR GXL DNA Polymerase (Takara Bio) and primers complementary to a common sequence of the lentivirus. For the first round of PCR, 35 reactions containing 1 μg gDNA were set up and amplified for 24 cycles. First, the PCR products were pooled, mixed, and subsequently subjected to second PCR. Then 5 mL of the mixed first PCR product were amplified for six cycles with index primers during the second round. PCR products were purified using the Agencourt AMPure XP beads (Beckman Coulter, Pasadena, CA, USA) and sequenced on the Illumina HiSeq2500 (Illumina, San Diego, CA, USA) according to the manufacturers' instructions. Primer sequences used in the study are listed in Supplementary Table 5.

**Generation of clonal knockout and complemented cell lines**. HAP1, AGS, or A549 cells were transfected with three different px459-sgRNA targeting the gene of interest (triple-target CRISPR) as described previously[67]. Next, the cells were selected using puromycin for 5 days. RPE-1 cells were transduced with a lentiviral vector generated by VectorBuilder®, followed by 7 days of antibiotic selection using blasticidin or neomycin. Clonal lines were established by limited dilution after selection. KO was confirmed through Sanger sequencing or Western blotting.

Lentiviral transduction was performed to generate stable cell lines complementarily expressing a selected gene of interest under a CMV promoter. Lentivirus was produced by transfecting HEK293T cells with CSII plasmids, pCMVR8.74 (a gift from Dider Trono and Yasuo Ariumi; #22036, Addgene) and pCMV-VSV-G (a gift from Bob Weinberg; #8454, Addgene). KO cells were transduced with the lentivirus encoding the gene of interest. Transduced cell populations were selected with blasticidin and the transgene expression was confirmed by immunoblotting.

**Heparinase treatment**. Cells were washed twice with PBS and incubated with 2 U/mL heparinase II diluted in PBS for 1 h at 37 °C. The cells were washed twice again with PBS and then subjected to subsequent experiments.

**HepS staining**. Cells were harvested using the TrypLE Select Enzyme (Sigma-Aldrich), washed with PBS, and subsequently incubated with monoclonal anti-HepS antibody diluted in PBS + 2% FBS. After 1 h of incubation at 4 °C, the cells were washed with PBS + 2% FBS and incubated with Alexa 488-conjugated secondary antibody. The cells were again washed twice with PBS + 2% FBS. Fluorescence was measured using the FACS Canto2 (BD Biosciences, Freemont, CA, USA) and analyzed using FlowJo software (FlowJo Inc, Ashland, OR, USA).

**Cell viability assay**. Cell viability was measured using the Cell Titer 96 AQueois One Solution reagent (Promega, Madison, WI, USA). In brief, wells containing 100 μL of the medium in 96-well plates were supplemented with 20 μL of 3-(4,5-Dimethylthiazol-2-yl)-5-(3-carboxymethoxyphenyl)-2-(4-sulfophenyl)-2H-tetrazolium, inner salt (MTS) reagent, followed by 2 h of incubation at 37 °C. Then, the absorbance was recorded at 490 nm using a 96-well plate reader (Sunrise Basic; Tecan Japan, Kawasaki, Japan).

**Progeny virus titer measurement**. Cells were infected with HSV-1 at an MOI of 1. After 24 h of incubation at 37 °C, the cells were collected with the supernatant and stored immediately at −80 °C. The samples were frozen and thawed thrice before titration. On the next day, 100 μL of the samples serially diluted in PBS were inoculated on pre-seeded Vero cells, then incubated at 37 °C for 1 h adsorption. After adsorption, 1 mL fresh medium containing 5% CS was added, and the cells were incubated for 48 h. These cells were fixed with 10% formaldehyde and stained with 0.5% crystal violet. Plaques were observed and counted.

**Virus binding assay**. For the virus binding assay, $2 \times 10^5$ cells were seeded into 24-well plates. On the next day, the medium was replaced with fresh medium containing 5 μg/mL of surfen hydrate or DMSO, followed by incubation for 1 h at 37 °C. The cells were inoculated with viruses in 100 μL ice-cold PBS. After 1 h of adsorption on an ice bath, the cells were washed with ice-cold PBS three times and collected for DNA extraction with the DNeasy Blood and Tissue Kit (QIAGEN) or RNA extraction with the NucleoSpin RNA Plus (Takara Bio). The HSV-1 and 2 viral DNA levels were measured via qPCR as previously described[68]. The HBV DNA and cDNA of HCoV OC43 were measured and normalized by *GAPDH* using the following primers: For HBV, 5′-GTG TCT GCG GCG TTT TAT CA-3′ and 5′-GAC AAA CGG GCA ACA TAC CTT-3′; for HCoV OC43, 5′-CGA TGA GGC TAT TCC GAC TAG GT-3′ and 5′-CCT TCC TGA GCC TTC AAT ATA GTA ACC-3′; for GAPDH, 5′-TGC ACC ACC AAC TGC TAG C-3′ and 5′-GGC ATG GAC TGT GGT CAT GAG-3′.

**Cell-to-cell spreading assay**. WT RPE-1 cell monolayers were exposed to HSV-1 at an MOI of 50 for 1 h at 37 °C to enable virus entry. Then, the cells were washed twice with PBS, followed by treatment with 40 mM of citrate buffer (pH 3.0) for 1 min to deactivate the extracellular residual viruses. The cells were again washed twice with PBS and collected to inoculate onto 90% confluent monolayers of

uninfected RPE-1 clones. Approximately 30 cells/well were inoculated with infection media (IMDM + 1% FBS) containing 5 mg/mL of pooled human IgG, which was previously reported to be a sufficient concentration to neutralize nearly all virions per $1 \times 10^6$ PFU[69]. At 72 hpi, the cells were fixed for 10 min in 10% formaldehyde and stained with 0.5% crystal violet. The number of plaques was counted manually by direct observation, and the area of plaques was analyzed using ImageJ Fiji software[70].

**Quantification of HepS chains**. Glycosaminoglycans (GAGs) from cultured cells were prepared as described previously[71] with slight modifications. Briefly, the cells were homogenized in ice-cold acetone at least three times, and air-dried. The dried materials (correspond to acetone powders) were exhaustively digested with heat-activated actinase E (10% [w/v] of dried materials) in 0.1 M borate buffer, pH 8.0, containing 10 mM $CaCl_2$, at 55 °C for 48 h. The digest was treated with 5% tri-chloroacetic acid, and the resultant acid-soluble fraction was adjusted to contain 80% ethanol. The resultant precipitate was dissolved in water and subjected to gel filtration on a PD-10 column (Cytiva), using water as an eluent. The flow-through fractions were collected, evaporated to dryness, and dissolved in water (crude GAG fractions).

An aliquot of the crude GAG sample was digested with a mixture of heparinase (0.5 mIU, Seikagaku, Tokyo, Japan) and heparitinase (0.5 mIU, Seikagaku) in 20 mM sodium acetate, pH 7.0, containing 2 mM calcium acetate at 37 °C for 4 h. The digests were derivatized with fluorophore 2-aminobenzamide and analyzed by anion-exchange high-performance liquid chromatography on a PA-G column (YMC, Kyoto, Japan), as described previously[72]. The identification and quantification of the resulting disaccharides were achieved by comparison with authentic unsaturated HepS disaccharides (Seikagaku).

The HepS sulfation degree was calculated as the mol% of the total sulfated HepS disaccharides multiplied by their respective sulfate group numbers (1, 2, or 3).

**Statistics and reproducibility**. Data were processed using Microsoft Excel. All analyses were performed using two-tailed Student's $t$-test. $P < 0.05$ was considered statistically significant. Results are presented as means ± standard error of measurement (SEM) of at least three independent experiments. The sample sizes and number of replicates were described in the figure legends. The experiments were not randomized and the investigators were not blinded.

**Reporting summary**. Further information on research design is available in the Nature Research Reporting Summary linked to this article.

## Data availability
The full list of raw sequencing read counts from the genome-wide CRISPR screen is available in Supplementary Data 1. All NGS datasets have been deposited in the DNA Data Bank of Japan (DDBJ; https://www.ddbj.nig.ac.jp/index-e.html) with the accession number DRA014278. Source data behinds the graphs in this article are provided in Supplementary Data 2. Uncropped blots are shown in Supplementary Fig. 6. All materials and other data are available from the corresponding authors on reasonable request.

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

## Acknowledgements

The authors thank Christoph Seeger (Fox Chase Cancer Center, USA) for HepAD38 cells; Hiroyuki Miyoshi (RIKEN, Japan), Feng Zhang (Broad Institute, USA), Didier Trono (Ecole Polytechnique Federale de Lausanne, Switzerland), Yasuo Ariumi (Kumamoto University, Japan), and Bob Weinberg (Whitehead Institute for Biomedical Research, USA) for plasmids; Tetsuya Okajima and Yasuyuki Miyake (Nagoya University Graduate School of Medicine, Japan) for valuable discussion; Ken Sago, Daisuke Sasaki, Sho Suzuki, and Tomoko Kunogi (Nagoya University Graduate School of Medicine, Japan) for technical assistance; and the Division for Medical Research Engineering at the Nagoya University Graduate School of Medicine for FACS analysis. This work was supported in part by grants from the Japan Society for the Promotion of Science (JSPS) KAKENHI (https://www.jsps.go.jp) (Grant Numbers JP16H06231 to Y.S., JP19H04829 to Y.S., JP21K15448 to Y.S., JP20K06551 to T.Mikami, JP20H03386 to H.Kitagawa; and JP20H03493 to H.Kimura); the JST (https://www.jst.go.jp) PRESTO (Grant Number JPMJPR19H5) to Y.S.; the Japan Agency for Medical Research and Development (AMED, https://www.amed.go.jp) (JP19jk0210023 to Y.S., JP21wm035042 to Y.S., JP19ck0106517 to Y.O., and JP20wm0325012 to T.Murata); the Takeda Science Foundation (https://www.takeda-sci.or.jp) to Y.S., Y.O., and T.Murata; the Hori Sciences and Arts Foundation (https://www.hori-foundation.or.jp) to Y.S., T.Murata, and H.Kimura; the MSD Life Science Foundation (https://www.msd-life-science-foundation.or.jp) to Y.S.; the Aichi Health Promotion Foundation (https://ahpf.or.jp) to T.S.; and the Uehara Memorial Foundation (https://www.ueharazaidan.or.jp/) to H. Kimura; and the Chemo-Sero-Therapeutic Research Institute (https://www.kaketsuken.org) to H.Kimura. TS is supported by the Takeda Science Foundation scholarship. The funders had no role in study design, data collection and analysis, decision to publish, or preparation of the manuscript.

## Author contributions

Y.S. and H.Kimura conceived the project. T.S., Y.S., Y.O., and H.Kimura designed the experiments. T.S., Y.S., and Y.O. performed experiments. T.Mikami and H.Kitagawa performed HPLC quantification. F.G., M.U., T.Murata, K.W., and T.Wakita provided resources. All authors analyzed data. T.S., Y.S., Y.O., and H.Kimura wrote the original draft. All authors read, discussed, and approved the manuscript.

## Competing interests

The authors declare no competing interests.
