## [Peer Review File · Communications Biology]

Reviewers' comments:

Reviewer #1 (Remarks to the Author):

In this manuscript the authors explore the role of 3'-phosphoadenosine 5'-phosphosulfate synthase 1 (PAPSS1) in HSV-1 infection and heparan sulfate (HepS) biosynthesis. They identified PAPSS1 as a novel factor of HSV-1 infection by a genome-wide CRISPR screen using near-haploid HAP1 cells. They also showed KO of PAPSS1 in HAP1 cells decreased HepS expression, and consequently diminished the binding of HSV-1 and several other HepS-dependent viruses. They further suggested PAPSS1-KO suppressed cell growth stimulation by HepS-mediated activation of the fibroblast growth factor signaling pathway.

To my opinion, the work contains several interesting findings, but altogether does not yet represent a fully coherent story that clearly provides mechanistic explanations of the observations reported and the following points have to be addressed.

Major concerns:

1. HAP1 cells are not common cells used for virology research, although it's good for gene knockout. It would be more convincing if the authors could confirm the relative results of viral infection with human immune cell lines, such as THP-1, before they claimed those genes, such as XYLT2, EXT2 and PAPSS1, affect HSV-1 infection.
2. The title of the paper shows the discovery of the article mainly focuses on a novel function of PAPSS1 in HepS biosynthesis, however, a large part of the results in the paper is about the effect of XYLT2, EXT2 and HepS on HSV-1 infection. It might be confusing to readers what the focus of the paper is. It would be better if the authors could clarify it by modifying either the title or the content of the paper.
3. Lines85-90 on Page 4, the authors infected cells with 0.1 MOI HSV-1 and had a second challenge with HSV-1 for the surviving cells to enrich the HSV-1 resistant cells. However, this cannot exclude the possibility that part of survivor cells they subjected to sequencing were not attacked by HSV-1 other than HSV-1 resistant cells. They should clarify it.
4. Fig 2E, to strengthen the conclusion that XYLT2 and EXT2 enhance the binding of HSV-1 on the cell surface through HepS expression, the authors should have extra treatments with surfen or heparinase in cells of XYLT2 KO+XYLT2, or in EXT2 KO+EXT2 to see whether surfen or heparinase will suppress the rescue of viral infection by the expression of XYLT2 or EXT2 in those cell lines. Same for the experiments in Fig3D-3F, and Fig4E-4H later on.

Minor points:

1. Lines95-97, and lines111-113, the authors found seven genes in the HepS biosynthesis pathway were markedly enriched after a genome-wide CRISPR screen, but only XYLT2 and EXT2 were selected for further study of their effect on viral infection and HepS expression. They should explain why they only selected these two genes.
2. Page5, line96, replace "XLYT2" with "XYLT2".
3. Line372, show the full name of MTS due to the first mention of this abbreviation.
4. Lines685-690, the authors should explain what the Mock in Fig2G is. Same for Fig4F and 4H later on.
5. Page25, Fig 1C, the authors should explain what the blue plots indicate.
6. Page25, Fig 1E, either "asterisks" or "n.s." is missing above the histogram of IRF2BPL KO#1.
7. Page26, Fig2B, please clarify which XYLT2 KO (KO#1 or KO#2?) or EXT2 KO (KO#1 or KO#2?) clone is shown here.
8. Page29, Fig5, it would be more convincing if the authors could provide PAPSS1 rescue clone data together with the data of ctrl and PAPSS1 KO clones they showed here.

Reviewer #2 (Remarks to the Author):

Herpes simplex viruses are ubiquitous pathogens that cause multiple life-threatening diseases in humans; therefore, there is much interest in identifying host factors as potential therapeutic targets to prevent infection. In this manuscript, the authors utilized genome-wide CRISPR screening to identify host factors for herpes simplex virus-1 (HSV-1) infection in HAP1 cells. Their CRISPR screen uncovered multiple genes known to be involved in HSV-1 infection, including the adhesion protein NECTIN1 and multiple enzymes involved in the biosynthesis of heparan sulfate, thus validating the screening approach. Previous studies have shown that cell surface heparan sulfate is essential for HSV-1 attachment and entry to human cells (Shukla et al. Cell 1999; Trybala et al J Virol 2020; Shukla et al. J Clin Invest 2001; O'Donnell et al. Virol Sin 2009); therefore, the novelty of this study is limited. The main hit focused on in the study, PAPSS1, is known as the synthase for PAPS, which is essential for sulfation of heparan sulfate (Fudu et al. Biochem J. 2002; Dejima et al JBC 2006; Xu et al Biochem J. 2005). Interestingly, the screen revealed enrichment of multiple genes not previously implicated in HSV-1 infection (IRF2BPL, VANGL2, MIR4647); however, none of these hits validated in follow-up infection experiments. Instead, heparan sulfate biosynthetic enzymes and genes involved in the synthesis of PAPS (PAPSS1) were validated in this study. Overall, the manuscript is very well written and easy to follow, presenting data of quality and appropriately using statistics. The experimental design and data analysis are rigorous, and the conclusions are justified by the data and will be of interest to the field. The key claims of the manuscript confirm previously known findings, thus limiting the impact of the work. I have outlined specific points which I suggest being addressed to improve the manuscript and strengthen the conclusions.

- The authors show that ablation of IRF2BPL and VANGL2 in HAP1 cells has no impact on HSV-1 infection. Please include data showing that these genes were in fact knocked out by western blotting and/or Sanger sequencing.
- The authors claim that ablation of PAPSS1 eliminated HepS expression in HAP1 cells. To make this claim, HepS amounts and/or composition should be measured in wildtype and PAPSS1 KO cells through metabolic [³⁵S]sulfate-labeling (Shworak, Meth Mol Biol 2001) and/or LC-MS analysis of HepS structure/composition (Volpi et al, Nat Prot 2014).
- Different infection protocols were used for the genome-wide screening assay (2 rounds of infection at low MOI over a long period of time) and the validation experiments (high MOI for ~48 hrs). It is possible this could have impacted the validation of novel hits, which showed no effect on HSV-1 infection in the generated knockout lines. Have the authors tried a lower MOI for validation infections? Please comment on why the infection protocol was altered and how this could have affected the validation of the screening hits in the discussion section.

Reviewer #3 (Remarks to the Author):

Communication Biology COMMSBIO-21-2592-T

In this short report, the authors reported that a genome-wide CRISPR screen for HSV-1 host factors revealed PAPSS1 as an essential factor for heparan sulfate biosynthesis and for HSV-1 infection. Using the CRISPR screen, the authors showed that several other host factors were also involved in the HepS biosynthesis pathway, in addition to PAPSS1, and were important for HSV-1 infection. The authors nicely characterized KO cells with XYLT2 and EXT2 and showed that disruption of the genes for these two factors led to a reduction of HepS synthesis and expression, decreased binding of HSV-1 to these cells, and inhibition of HSV-1 infection. Similarly, KO of PAPSS1 in cells led to a reduced HepS synthesis and expression, decreased binding of HSV-1 to these cells, and inhibition of HSV-1 infection and replication. The experiments were well conceived and clearly described. In general, the conclusions of the manuscripts are supported by the results. Several issues may need to be addressed before publication.

1. The important roles of HepS in HSV-1 attachment and entry have been well documented previously.

The novelty of the manuscript is that PAPSS1 is a host factor important for HSV-1 infection and is involved in HepS synthesis. The screen was set up to identify host factors that are required to support HSV-1 infection. It is reasonable to suggest that the screen should yield a collection of host factors that are important for many steps of the viral infection cycle (e.g. attachment and fusion steps of viral entry, decoating and trafficking of the viral capsid before its import to the nucleus, and initiation of viral transcription, etc.). It is surprising to see that only host factors that are involved in attachment and fusion steps have been identified from the screen. I do not believe that the authors imply in their study that only host factors involved in viral entry (e.g. HepS synthesis) are important for HSV-1 infection. Additional results and discussion may need to be included to address this concern. A possible explanation for their results is that the screen, which was set up to screen surviving cells after exposure to HSV-1, was only able to identify host factors blocking viral attachment and entry since HSV-1 attachment and entry could have triggered apoptosis, leading to cell death.

2. The relevance of the results from the experiments with the FGF2 (e.g. Figure 5) to the general conclusions of the manuscript is not clear. I would suggest the removal and deletion of this part of the study as it does not relate to HSV-1 infection.

3. I am little puzzled with the level of cell viability and reduction of viral titers in cells with KO of the identified factors. About 10-30% KO cells complementing with XYLT2, EXT2, and PAPSS1 remained resistant with HSV-1 infection and survived (Figure 2E and 4E). A rationale or explanation for this substantial level of viability may be needed. Perhaps this is because of the transfection efficiency. Additional experiments may be included in which enrichment of transfected cells will be carried out so higher percentage of transfected cells will be used in order to show that less than 10% cells exhibited viability. Experiments with cell populations with higher percentage of transfected cells will also yield more substantial difference in the viral titers.

Point-by-point Responses to Reviewer's comments:

We are very grateful for the careful review of our manuscript and the constructive comments provided. Below are our point-by-point responses (blue) to the Reviewers' comments (black,):

Responses to Reviewer #1

Reviewer #1 (Remarks to the Author):

In this manuscript the authors explore the role of 3' -phosphoadenosine 5' -phosphosulfate synthase 1 (PAPSS1) in HSV-1 infection and heparan sulfate (HepS) biosynthesis. They identified PAPSS1 as a novel factor of HSV-1 infection by a genome-wide CRISPR screen using near-haploid HAP1 cells. They also showed KO of PAPSS1 in HAP1 cells decreased HepS expression, and consequently diminished the binding of HSV-1 and several other HepS-dependent viruses. They further suggested PAPSS1-KO suppressed cell growth stimulation by HepS-mediated activation of the fibroblast growth factor signaling pathway.

To my opinion, the work contains several interesting findings, but altogether does not yet represent a fully coherent story that clearly provides mechanistic explanations of the observations reported and the following points have to be addressed.

Major concerns:

1. HAP1 cells are not common cells used for virology research, although it's good for gene knockout. It would be more convincing if the authors could confirm the relative results of viral infection with human immune cell lines, such as THP-1, before they claimed those genes, such as XYLT2, EXT2 and PAPSS1, affect HSV-1 infection.

We thank the Reviewer for their valuable comments. We have evaluated the basal expression levels of HepS in various immune cell lines and revealed that HepS expression was low in these cell lines, including THP-1 (Figure A1 below). Consistent with this, the efficiency of HSV-1 infection in these immune cells was very poor. Therefore, our findings proposed that the effect of XYLT2-, EXT2-, and PAPSS1-KOs on HepS expression in these cells would be limited.

Figure A1: Poor herpes simplex virus type 1 (HSV-1) infection efficiency in immune cells because of their low heparan sulfate (HepS) expression.

(A) HepS expression analysis by fluorescence-activated cell sorting. An anti-HepS antibody was used to stain wildtype THP-1, Akata, Raji, Daudi, and RPE-1 cells.

(B) Viability of wildtype THP-1, Akata, Raji, Daudi, and RPE-1 cells following HSV-1 infection. The cells were infected with HSV-1 at a multiplicity of infection of 10. Then, the viabilities were measured via the MTS assay at 48 hpi. The results are presented as means \pm SEM of three independent experiments.

Although HSV-1 has broad cell tropism, it is a neurotropic alphaherpesvirus that predominantly infects epithelial cells and neurons. Therefore, we obtained KO cells from RPE-1 cells (hTERT-immortalized retinal pigment epithelial cells) that is often used in HSV-1 studies. The RPE-1 KO clones of *XYLT2* and *EXT2* showed reduced HepS expression to the background level and resistance against HSV-1 infection, similar to the KOs in HAP1 cells (Figure 2H and

Supplementary Figure 3).

2. The title of the paper shows the discovery of the article mainly focuses on a novel function of PAPSS1 in HepS biosynthesis, however, a large part of the results in the paper is about the effect of XYLT2, EXT2 and HepS on HSV-1 infection. It might be confusing to readers what the focus of the paper is. It would be better if the authors could clarify it by modifying either the title or the content of the paper.

In the revised manuscript, we have added data regarding *PAPSS2*-KO and *PAPSS1/PAPSS2*-double KO RPE-1 cells to bring focus on the function of *PAPSS1* in HepS biosynthesis. Although the KO of *XYLT2* or *EXT2* showed reduced HepS expression to the background level in RPE-1 cells (Figure 2H and Supplementary Figure 3), the reduction of HepS levels in *PAPSS1*-KO was limited. Consistent with this finding, the *PAPSS1*-KO clones presented comparable resistance with that of the control clone against HSV-1 infection (Figure 5). Additionally, we obtained *PAPSS1*-KO clones in lung A549 cells and gastric AGS cells. The genetic ablation of *PAPSS1* slightly reduced HepS expression in A549 and drastically in AGS cells (Figure 5 and Supplementary Figure 3). Therefore, to address the difference in the *PAPSS1* dependency of HepS biosynthesis among these cell lines, we evaluated the expression of *PAPSS2*, another isoform of PAPS synthase, in humans (Figure 5). The double KO of *PAPSS1* and *PAPSS2* in RPE-1 cells reduced HepS expression to the background level and conferred robust resistance against HSV-1 infection (Figure 5). We also confirmed that the KO of only *PAPSS2* was not enough to render RPE-1 cells resistant to HSV-1 infection. Overall, our study suggested that for the cell lines that express not only *PAPSS1* but also *PAPSS2* affects HSV-1 infection through HepS biosynthesis.

3. Lines 85-90 on Page 4, the authors infected cells with 0.1 MOI HSV-1 and had a second challenge with HSV-1 for the surviving cells to enrich the HSV-1 resistant cells. However, this cannot exclude the possibility that part of survivor cells they subjected to sequencing were not attacked by HSV-1 other than HSV-1 resistant cells. They should clarify it.

Since the progeny viruses were released into the medium from the infected cells two weeks after the selection period and after the second challenge, we considered that the cells were practically exposed to HSV-1 with a MOI higher

than 0.1. Therefore, to ensure that all the CRISPR library cells were exposed to HSV-1, the same number of parental HAP1/Cas9 cells were cultured and infected with HSV-1 as a control. We confirmed no parental survival cells post-infection. This clarification has been added to the Methods section of the revised manuscript (lines 356-9).

4. Fig 2E, to strengthen the conclusion that XYLT2 and EXT2 enhance the binding of HSV-1 on the cell surface through HepS expression, the authors should have extra treatments with surfen or heparinase in cells of XYLT2 KO+XYLT2, or in EXT2 KO+EXT2 to see whether surfen or heparinase will suppress the rescue of viral infection by the expression of XYLT2 or EXT2 in those cell lines. Same for the experiments in Fig3D-3F, and Fig4E-4H later on. As per your suggestion, we performed additional treatments with surfen in the rescued cells (Figure 2E–G and Figure 4F–H).

In the cell spreading assay, as shown in Figures 3 D–F, the surfen or heparinase treatments of WT RPE-1 cells showed limited or no effect on plaque size (see below Figure A2). Therefore, we did not perform additional treatments with surfen or heparinase in *EXT2-KO + EXT2* RPE-1 cells.

Specifically, heparinase had to be removed during the incubation period because a long heparinase treatment resulted in the cells detaching from the culture dish, causing only a transient effect of heparinase after treatments. A possible explanation for the limited effect of surfen treatment on cell-to-cell infection is that surfen could not permeate the tight junctions formed by the RPE-1 cells and was effective only after the first contact with the inoculating infected cells and recipient RPE-1 cells.

Figure A2: Limited effect of surfen or heparinase treatment on cell-to-cell infection.

(A and B) Herpes simplex virus type 1 (HSV-1) cell-to-cell spreading assay. Wildtype human retinal epithelial (RPE-1) cells were treated with heparinase or were left untreated and were inoculated with 30 cells/well of wildtype RPE-1, pre-infected with HSV-1. Then, they were incubated in a media containing 5 mg/mL pooled human IgG and surfen or DMSO (A). Subsequently, the areas of 10 individual plaques were measured (B). The results are presented as means \pm SEM of three independent experiments and are shown as the relative size of plaques (B). Asterisks, $p < 0.05$; n.s., not significant.

Minor points:

1. Lines95-97, and lines111-113, the authors found seven genes in the HepS biosynthesis pathway were markedly enriched after a genome-wide CRISPR screen, but only XYLT2 and EXT2 were selected for further study of their effect on viral infection and HepS expression. They should explain why they only selected these two genes.

We selected *XYLT2* and *EXT2* because these two genes are the initiator and the finisher of the HepS backbone formation, respectively. We have added the corresponding clarification to the revised text (lines 102-4).

2. Page5, line96, replace “XLYT2” with “XYLT2.”

Thank you for the careful observation. We have made this correction in the revised manuscript (line 90).

3. Line372, show the full name of MTS due to the first mention of this abbreviation.

MTS's full name has been written at first mention in the text (Lines 405-6).

4. Lines685-690, the authors should explain what the Mock in Fig2G is. Same for Fig4F and 4H later on.

Information regarding Mock has been added in all the figure legends of the revised manuscript (lines 743-4, 787-8, and 797-8).

5. Page25, Fig 1C, the authors should explain what the blue plots indicate.

The blue plots in the original manuscript indicated that the genes were above the red lines and were not categorized as black or green plots. To avoid confusion, the blue color has been replaced with a base gray color in the revised manuscript.

6. Page25, Fig 1E, either “asterisks” or “n.s.” is missing above the histogram of IRF2BPL KO#1.

We have added the missing details to the result of the statistical analysis above the histogram of *IRF2BPL*-KO#1 in Figure 1E.

7. Page26, Fig2B, please clarify which *XYLT2* KO (KO#1 or KO#2?) or *EXT2* KO (KO#1 or KO#2?) clone is shown here.

The clone names used in Figure 2B have been clarified in the revised manuscript.

8. Page29, Fig5, it would be more convincing if the authors could provide *PAPSS1* rescue clone data together with the data of ctrl and *PAPSS1* KO clones they showed here.

In line with Reviewer #3's recommendation, we have deleted the experiments with *FGF2* in the revised manuscript.

Responses to Reviewer #2

Reviewer #2 (Remarks to the Author):

Herpes simplex viruses are ubiquitous pathogens that cause multiple life-threatening diseases in humans; therefore, there is much interest in identifying host factors as potential therapeutic targets to prevent infection. In this manuscript, the authors utilized genome-wide CRISPR screening to identify host factors for herpes simplex virus-1 (HSV-1) infection in HAP1 cells. Their CRISPR screen uncovered multiple genes known to be involved in HSV-1 infection, including the adhesion protein NECTIN1 and multiple enzymes involved in the biosynthesis of heparan sulfate, thus validating the screening approach. Previous studies have shown that cell surface heparan sulfate is essential for HSV-1 attachment and entry to human cells (Shukla et al. Cell 1999; Trybala et al J Virol 2020; Shukla et al. J Clin Invest 2001; O'Donnell et al.

Virol Sin 2009); therefore, the novelty of this study is limited. The main hit focused on in the study, *PAPSS1*, is known as the synthase for PAPS, which is essential for sulfation of heparan sulfate (Fudu et al. *Biochem J.* 2002; Dejima et al *JBC* 2006; Xu et al *Biochem J.* 2005). Interestingly, the screen revealed enrichment of multiple genes not previously implicated in HSV-1 infection (*IRF2BPL*, *VANGL2*, *MIR4647*); however, none of these hits validated in follow-up infection experiments. Instead, heparan sulfate biosynthetic enzymes and genes involved in the synthesis of PAPS (*PAPSS1*) were validated in this study. Overall, the manuscript is very well written and easy to follow, presenting data of quality and appropriately using statistics. The experimental design and data analysis are rigorous, and the conclusions are justified by the data and will be of interest to the field. The key claims of the manuscript confirm previously known findings, thus limiting the impact of the work. I have outlined specific points which I suggest being addressed to improve the manuscript and strengthen the conclusions.

- *The authors show that ablation of IRF2BPL and VANGL2 in HAP1 cells has no impact on HSV-1 infection. Please include data showing that these genes were in fact knocked out by western blotting and/or Sanger sequencing.*

We thank the Reviewer for their important comments. We have added western blotting data confirming the knockout of *IRF2BPL* and *VANGL2* to the revised manuscript (Supplementary Figure 1A).

- *The authors claim that ablation of PAPSS1 eliminated HepS expression in HAP1 cells. To make this claim, HepS amounts and/or composition should be measured in wildtype and PAPSS1 KO cells through metabolic [³⁵S]sulfate-labeling (Shworak, *Meth Mol Biol* 2001) and/or LC-MS analysis of HepS structure/composition (Volpi et al, *Nat Prot* 2014).*

We performed disaccharides analysis of HepS through HPLC using wildtype, nontargeting control, and *PAPSS1*-KO cells. The total HepS disaccharide levels were reduced by approximately 30% in *PAPSS1*-KO cells compared with the wildtype or nontargeting control cells. Additionally, the HepS sulfation degree drastically dropped from 80% to 4% after the ablation of *PAPSS1* (Figure 4E and Supplementary Figure 4), suggesting the importance of *PAPSS1* in HepS biosynthesis.

• *Different infection protocols were used for the genome-wide screening assay (2 rounds of infection at low MOI over a long period of time) and the validation experiments (high MOI for ~48 hrs). It is possible this could have impacted the validation of novel hits, which showed no effect on HSV-1 infection in the generated knockout lines. Have the authors tried a lower MOI for validation infections? Please comment on why the infection protocol was altered and how this could have affected the validation of the screening hits in the discussion section.*

Although we also validated KO cells with a lower MOI and longer periods (MOI = 0.1 for 72 hpi), no significant resistance was observed in the *IRP2BPL*- and *VANGL2*-KO clones (Supplementary Figure 1B). We have added the corresponding information in the revised text (lines 110-1). Subsequently, we used the infection protocol to validate our screen hits, leading to a clear difference between the control and host factors for HSV-1 infections.

Responses to Reviewer #3

Reviewer #3 (Remarks to the Author):

In this short report, the authors reported that a genome-wide CRISPR screen for HSV-1 host factors revealed PAPSS1 as an essential factor for heparan sulfate biosynthesis and for HSV-1 infection. Using the CRISPR screen, the authors showed that several other host factors were also involved in the HepS biosynthesis pathway, in addition to PAPSS1, and were important for HSV-1 infection. The authors nicely characterized KO cells with XYLT2 and EXT2 and showed that disruption of the genes for these two factors led to a reduction of HepS synthesis and expression, decreased binding of HSV-1 to these cells, and inhibition of HSV-1 infection. Similarly, KO of PAPSS1 in cells led to a reduced HepS synthesis and expression, decreased binding of HSV-1 to these cells, and inhibition of HSV-1 infection and replication. The experiments were well conceived and clearly described. In general, the conclusions of the manuscripts are supported by the results. Several issues may need to be addressed before publication.

1. The important roles of HepS in HSV-1 attachment and entry have been well documented previously. The novelty of the manuscript is that PAPSS1 is a host factor important for HSV-1 infection and is involved in HepS synthesis. The

screen was set up to identify host factors that are required to support HSV-1 infection. It is reasonable to suggest that the screen should yield a collection of host factors that are important for many steps of the viral infection cycle (e.g., attachment and fusion steps of viral entry, decoating and trafficking of the viral capsid before its import to the nucleus, and initiation of viral transcription, etc.). It is surprising to see that only host factors that are involved in attachment and fusion steps have been identified from the screen. I do not believe that the authors imply in their study that only host factors involved in viral entry (e.g., HepS synthesis) are important for HSV-1 infection. Additional results and discussion may need to be included to address this concern. A possible explanation for their results is that the screen, which was set up to screen surviving cells after exposure to HSV-1, was only able to identify host factors blocking viral attachment and entry since HSV-1 attachment and entry could have triggered apoptosis, leading to cell death.

We thank the Reviewer for their constructive comments. The host factors enriched in our CRISPR screen were related to HSV-1 viral attachment or entry. However, no host factor involved in the later stages of the infection was isolated in our screen. A possible reason for this bias is that although the cells lacking host factors can engage in the later stages of infection and survive the HSV-1 infection, they did not propagate sufficiently or undergo apoptosis so as to harbor viruses during the selection period (~ 4weeks). We have discussed this point in the revised manuscript (lines 240-6).

2. The relevance of the results from the experiments with the FGF2 (e.g., Figure 5) to the general conclusions of the manuscript is not clear. I would suggest the removal and deletion of this part of the study as it does not relate to HSV-1 infection.

Following the Reviewer's comment, we have deleted the experiments with FGF2 from the revised manuscript.

3. I am little puzzled with the level of cell viability and reduction of viral titers in cells with KO of the identified factors. About 10-30% KO cells complementing with XYLT2, EXT2, and PAPPS1 remained resistant with HSV-1 infection and survived (Figure 2E and 4E). A rationale or explanation for this substantial level of viability may be needed. Perhaps this is because of the transfection efficiency. Additional experiments may be included in which enrichment of transfected cells

will be carried out so higher percentage of transfected cells will be used in order to show that less than 10% cells exhibited viability. Experiments with cell populations with higher percentage of transfected cells will also yield more substantial difference in the viral titers.

In the original manuscript, we performed experiments using the complement cells without cloning after transfection and antibiotic selection. However, as you have rightly highlighted, the persistent resistance of the complement cells was due to their transfection efficiencies. Therefore, to remove this possibility, we isolated the complement clones through limited dilution/antibiotic selection and included the results of the assessments that were performed using these clones in the revised manuscript (Figures 2 and 4).

REVIEWERS' COMMENTS:

Reviewer #1 (Remarks to the Author):

The revised manuscript has taken my comments into account, and improved substantially. To my opinion, it can be considered for acceptance.

Reviewer #2 (Remarks to the Author):

The revised manuscript includes important new data that was missing from the previous version and addresses most of the key issues raised in the previous review of the manuscript. Before publication, the authors need to address the minor issue listed below:

1) Lines 235-239 mention that XYLT2 and EXT2 are "the enzymes that initiate or finish HepS backbone formation". This is not necessarily correct. Please correct to state that XYLT2 is a glycosyltransferase that initiates the biosynthesis of glycosaminoglycan chains (both HS and CS/DS) in proteoglycans, and EXT2 is involved in heparan sulfate chain elongation. Proper references should be included as well.

Reviewer #3 (Remarks to the Author):

Communication Biology COMMSBIO-21-2592A

In this revised report, the authors reported that a genome-wide CRISPR screen for HSV-1 host factors revealed PAPSS1 as an essential factor for heparan sulfate biosynthesis. Using the CRISPR screen, the authors showed that several other host factors were also involved in the HepS biosynthesis pathway, in addition to PAPSS1, and were important for HSV-1 infection.

Overall, the experiments were well conceived and clearly described. The conclusions of the manuscripts are supported by the results. However, the novelty of the manuscript remain very limited, since extensive studies in the past have shown that heparan sulfate is essential for HSV-1 entry and infection. The reviewers addressed most of the previous reviewers' comments. A minor issue may need to be further addressed before publication.

Previous reviewers raised concerns about the bias of the selection screen for identification of only the host factors required for HSV-1 entry or early steps of viral infection cycle. This is because HSV-1 is known to be cytopathic and cytotoxic, and it is well known that infection with UV-inactivated HSV-1 could lead to cytopathic and cytotoxic effects, even in the absence of viral transcription and genome replication. The authors attempted to address this issue in the revised manuscript (line 240-246) and unfortunately, their revised statements are still confusing. The authors are encouraged to provide further clarification on this issue.

Point-by-point Responses to Reviewer's comments:

We are very grateful for the careful review of our manuscript and the constructive comments provided. Below are our point-by-point responses (blue) to the Reviewers' comments (black,):

Responses to Reviewer #1

Reviewer #1 (Remarks to the Author):

The revised manuscript has taken my comments into account, and improved substantially. To my opinion, it can be considered for acceptance.

Thank you for your kind consideration.

Responses to Reviewer #2

Reviewer #2 (Remarks to the Author):

The revised manuscript includes important new data that was missing from the previous version and addresses most of the key issues raised in the previous review of the manuscript. Before publication, the authors need to address the minor issue listed below: 1) Lines 235-239 mention that XYLT2 and EXT2 are "the enzymes that initiate or finish HepS backbone formation". This is not necessarily correct. Please correct to state that XYLT2 is a glycosyltransferase that initiates the biosynthesis of glycosaminoglycan chains (both HS and CS/DS) in proteoglycans, and EXT2 is involved in heparan sulfate chain elongation. Proper references should be included as well.

Thank you for your comments and suggestions. We corrected the sentence explaining XYLT2 and EXT2, in line with your suggestion (Lines 127-129 and 261-264).

Responses to Reviewer #3

Reviewer #3 (Remarks to the Author):

In this revised report, the authors reported that a genome-wide CRISPR screen for HSV-1 host factors revealed PAPSS1 as an essential factor for heparan sulfate biosynthesis. Using the CRISPR screen, the authors showed that several other host factors were also involved in the HepS biosynthesis pathway, in addition to PAPSS1, and were important for HSV-1 infection.

Overall, the experiments were well conceived and clearly described. The conclusions of the manuscripts are supported by the results. However, the novelty of the manuscript remain very limited, since extensive studies in the past have shown that heparan sulfate is essential for HSV-1 entry and infection. The reviewers addressed most of the previous reviewers' comments. A minor issue may need to be further addressed before publication.

Previous reviewers raised concerns about the bias of the selection screen for identification of only the host factors required for HSV-1 entry or early steps of viral infection cycle. This is because HSV-1 is known to be cytopathic and cytotoxic, and it is well known that infection with UV-inactivated HSV-1 could lead to cytopathic and cytotoxic effects, even in the absence of viral transcription and genome replication. The authors attempted to address this issue in the revised manuscript (line 240-246) and unfortunately, their revised statements are still confusing. The authors are encouraged to provide further clarification on this issue.

Thank you for your comments and suggestions. We revised the sentences describing the reason why isolated host factors of our screen were involved in HSV-1 entry or early steps of viral infection cycle (Lines 267-275).